# Influence of lighting on sleep behaviour, circadian rhythm and spontaneous blink rate in stabled riding school horses (*Equus caballus*)

**Linda Greening[1], Eilis Harkin[2], Panoraia Kyriazopoulou**[3], **Zoe Heppelthwaite[4], Francesca Aragona[2]¤, John A. Browne[2], Andrew Hemmings[4], Jane M. Williams[1], Barbara A. Murphy**[2]*

1 Equine Department, Hartpury University, Gloucester, Gloucestershire, United Kingdom, 2 School of Agriculture and Food Science, University College Dublin, Belfield, Dublin, Ireland, 3 Equilume Ltd, Naas, Co. Kildare, Ireland, 4 Royal Agricultural University, Cirencester, Gloucestershire, United Kingdom

¤ Current address: Department of Veterinary Sciences, University of Messina, Messina, Italy
* barbara.murphy@ucd.ie

## Abstract

Modern horse husbandry involves significant time spent indoors, often in suboptimal lighting conditions and with frequent night-time disturbances by humans for management purposes. The aim of this study was to investigate the influence of a customised light-emitting diode (LED) lighting system and a standard fluorescent lighting fixture on equine sleep behaviours, circadian rhythmicity and spontaneous blink rates in horses. Ten riding school horses experienced two stable lighting conditions for four weeks each in a cross-over study running from January to March, 2023. The treatment lighting consisted of an LED system that provided timed, blue-enriched white polychromatic light by day and dim red light at night, and control lighting was a fluorescent tube that was turned on and off manually morning and evening. During week 4 of each experimental period, spontaneous blink rate was recorded twice for 30 min, behaviour of horses in their stables was recorded continuously for 72 h, and hair samples for circadian clock gene analysis were collected at 4-h intervals for 52 h. No differences were detected for total sleep, lateral or sternal recumbency, wakefulness, standing, standing sleep, or spontaneous blink rate (P > 0.05), between lighting conditions. The lighting period (Day versus Night) influenced total sleep (P < 0.01), total recumbency (P < 0.01), wakefulness (P < 0.01), and standing sleep (P < 0.05) in both conditions. For the treatment condition only, higher wakefulness was recorded during Day (P < 0.05). An overall effect of time for clock genes *PER2* and *DBP* was detected (P < 0.01), but there was no effect of treatment, or time by treatment interaction. Cosinor analysis detected significant 24-h rhythmicity for *PER2* and *DBP* (P < 0.01) in both lighting conditions. Results imply that dim red light at night does not negatively impact normal sleep patterns or circadian rhythmicity, and provide evidence supporting further research to better understand the role of blue-enriched LED light at promoting increased wakefulness during daytime in stabled horses.

**Data availability statement:** The data underlying the results presented in the study are available from the following online repository: https://github.com/EquineResearch/Sleep-Behaviour_2024.

**Funding:** Funding for this work was provided by a grant award from the Morris Animal Foundation (https://www.morrisanimalfoundation.org/) withGrant ID and Title: D22EQ-514/ Seeing the light to JMW, BAM and LG. The funders did not play any role in the study design, data collection and analysis, decision to publish, or preparation of the manuscript.

**Competing interests:** I have read the journal's policy and the authors of this manuscript have the following competing interests: BAM is the Founder of Equilume Ltd., a spin-out company deriving from her research program as associate professor at University College Dublin and is a member of the company's Board of Directors. BAM is a shareholder in Equilume Ltd. PK is an employee of Equilume Ltd. The treatment lighting condition in the present study comprised an Equilume Stable Light and is a commercially available product. This does not alter our adherence to PLOS ONE policies on sharing data and materials.

## Introduction

Equestrianism is popular worldwide, with millions of horses and riders participating in competitive horse sports and non-competitive leisure riding [1]. Horse riders, owners and trainers are responsible for the management of their horses, with a duty of care to engage in practices which optimise equine health and welfare [2]. Horses (*Equus Caballus*) used in equestrian sport are often stabled as part of their normal management regimes, with up to 36% stabled all year round in the UK according to the recent British Equestrian Trade Association survey [3], despite evidence that stabling can compromise equine welfare by increasing stereotypic and abnormal repetitive behaviours [4]. Similarly, stable lighting regimes as well as bedding depth are known to influence sleep behaviour and quality, with less time spent in recumbent sleep stages on sub-optimal bedding [5] and sleep disturbances associated with nighttime light exposure [6]. Sleep is critical to the well-being of all animals and is co-regulated via circadian and homeostatic mechanisms [7]. It is affected by a range of environmental factors in horses [8], whereas light is the primary stimulus affecting sleep in humans [9,10]. However, the effects of disrupted sleep on welfare are currently poorly understood by both animal welfare scientists [11] and horse owners alike [12].

The circadian rhythm is an endogenous 24-hour cycle regulating physiology and behaviour entrained by external stimuli [13], primarily by the daily alternating periods of light and dark [14]. The retina receives light signals and translates them into neural signals that travel along the retino-hypothalamic tract to the suprachiasmatic nucleus (SCN) [15]. The SCN, the master mammalian circadian clock located in the hypothalamus [16], synchronises the body's internal functions with the external environment via neural and hormonal signals that in turn signal time-of-day messages to peripheral clocks located throughout the body [17,18]. Circadian synchronisation drives self-sustained mammalian circadian rhythms of rest-activity cycles, hormone secretion, cardiovascular activity, metabolism, immune function, alertness, and musculoskeletal performance [17–20]. Tissue-specific gene expression patterns align organ function with the environment [21], such that circadian measurements can be taken from almost all peripheral tissues [22]. The use of hair follicle cells provides a practical, non-invasive means of assessing the internal circadian clock phase, and the strength of circadian synchronisation with the environment [19,23]. Cyclical expression of clock genes *NR1D2* (Nuclear receptor subfamily 1, group D, member 2), *Per2* (Period homolog 2), and the clock-controlled gene *Dbp* (D-site of albumin promoter binding protein) have successfully been confirmed in both human and horse hair follicles [23,24].

As a primary mediator of the circadian signal throughout the body, the hormone melatonin is also responsible for regulating the sleep-wake cycle [25]. Light signals processed by the SCN signal the pineal gland to produce melatonin during the hours of darkness [26]. Melatonin rhythmically fluctuates between low circulating levels by day and high levels during darkness hours [27]. As well as its important role in transmitting and maintaining circadian and circannual messages, melatonin is an important health regulator, functioning to optimize night-time rest [28] and immune function

[29]. Therefore, consistent sleep-wake cycles and healthy functioning of the circadian system requires the absence of light at night to facilitate the normal rise in melatonin production [30].

The circadian system and melatonin suppression are not equally affected by all types of light. The intrinsically photo-sensitive retinal ganglion cells within the retina that are responsible for regulating the master circadian clock are max-imally stimulated by short wavelength blue light (465–485 nm) [31]. Circadian disruption from exposure to sub-optimal light spectra and intensities by day, as well as erratic exposure to light at night, is responsible for a wide array of human health conditions [32,33]. Intensely managed horses often spend a high proportion of the 24 h period stabled and can miss optimal daytime light intensities or the natural fluctuations in the environmental light-dark cycle. Anecdotally, incan-descent bulbs and fluorescent tubes are more commonly used on equine establishments and predominantly emit light with peaks within the longer wavelength, orange/red region of the spectrum, that are not ideal for optimizing circadian rhythms [34]. Previous studies in horses showed that low-intensity blue light (465 nm) optimally suppressed melatonin [35] and stable lighting incorporating blue-enriched polychromatic light by day provided a stronger entrainment cue to the circadian system [36].

Furthermore, equine management frequently necessitates interaction with horses at night, resulting in abrupt expo-sures to light at inappropriate times in the 24 h cycle. Exposure to bright light at night very strongly suppresses melatonin secretion in humans and other animals [28,37]. A similar acute suppressive affect has been observed for horses [35], with evidence of disruption to the sleep-wake cycle for stabled horses caused by different lighting regimes [5,36] and an immediate drop in plasma melatonin concentrations was shown to accompany the switching on of stable lights at night [38]. Conversely, low intensity red light does not disrupt the nightly melatonin rise in horses [39] and human studies have shown that red light, with wavelengths above 620 nm, can be used to permit night-time visibility, yet minimize circadian disruption and maintain melatonin production [40,41]. In rodents, white light intensities as low as 5 lux have been shown to disrupt the sleep-wake cycle [42], whereas <10 lux of red light facilitated normal rest-activity behaviour and metabolic function [43,44]. These findings led to the development of a stable lighting system incorporating blue-enriched polychro-matic white light by day and red light at night [36], but the impact of this lighting system on sleep behaviour in horses has not yet been investigated. Finally, perturbations in sleep homeostasis have previously been linked to activation of stress pathways such as the hypothalamic-pituitary-adrenal axis (HPAA), in a range of mammalian species (see [45] for recent review). Moreover, cortisol as a primary endocrine mediator of the stress response has been positively correlated with spontaneous blink rate (SBR) in the horse [46]. In this regard, SBR has been used as a robust stress indicator in horses subjected to sudden arrival of a novel object [47] and sham clipping [48]. Therefore, SBR was utilised in this investigation, to gauge stress levels in animals subjected to both LED and fluorescent tube lighting systems.

Due to the meaningful impact of light on daily biological functioning, it is increasingly important to understand the con-sequences of lighting regimes on equine circadian regulation, sleep and arousal levels. This study therefore compared the impact of two lighting systems on sleep behaviour, circadian rhythm and spontaneous blink rates in stabled riding school horses. We hypothesized that these parameters would be impacted in horses under LED lighting providing timed exposure to blue-enriched light by day and dim red light at night compared to a standard manual lighting regime utilising fluorescent light fixtures.

## Methods

A crossover study design was used to investigate circadian clock gene expression in hair follicles, behavioural parameters related to sleep and wakefulness, and spontaneous blink rates in stabled horses under two lighting conditions: a custom-ised Light-emitting diode (LED) lighting system (Equilume Stable Light; Equilume Ltd, Co. Kildare, Ireland; www.equilume.com) (Treatment) and a standard fluorescent lighting fixture (Control). The experiment involved behavioural observations of video footage, collection of hair follicle samples, as previously described [23,36], and observations of spontaneous blink rate (SBR). All procedures were described under USDA Category B. The study was conducted in an equestrian stabling

facility at Hartpury University, Gloucestershire, United Kingdom, latitude 51.91012°, longitude −2.30609°. The Ethics Committee of Hartpury University granted ethical approval for the study with grant approval number ETHICS2021-63.

## Animals

Ten horses aged between five and seventeen years of mixed breed (riding horse/cob types), height, and sex (n = 7 geldings; n = 3 mares) were recruited through convenience sampling from a population of riding horses resident at Hartpury Equestrian Centre. All horses were acclimatised to their individual stables, measuring 3.6 m x 3.6 m and arranged in an American barn with a walkway in the centre, for one month prior to study commencement. The stables were partitioned by wooden panels with metal bars at the front and an open-top stable door facing into the central aisle. All horses were bedded on wood shavings (depth ~10 cm) covering approximately half of the stable floor, with rubber matting beneath covering the full floor. The stables were cleaned each morning between 07:00–08:00 and intermittently throughout the day until 19:00. Horses were rugged overnight according to individual needs. Horses had ad-lib access to water provided in a bucket and refreshed regularly throughout the day. One of the horses was fed twice daily with 2.5 kg of concentrated feed (Baileys Horse Feeds) in addition to ad-lib hay. The diets of the remaining nine horses consisted entirely of hay, which was provided ad-lib, so they did not exhaust their supply. No horses displayed signs of stereotypy, and all horses were deemed fit for exercise. The horses were stabled continuously but had a daily exercise and management period during which they were out of the stable for one to two and a half hours. The exercise timing (period out of the stable) occurred between 09:00 and 18:00 and varied per horse and day throughout the study.

## Study design

Horses were randomly allocated to individual stables by the yard manager and then assigned to one of two groups A (mean age = 11.2 ± 4.1 years) or B (mean age = 12 ± 5 years) according to their location in the barn (Fig 1).

Each individual stable had both a Treatment and Control lighting fixture fitted, as follows: Treatment lighting comprised an Equilume® Stable Light (Equilume Ltd, Kildare, Ireland) that was secured 3.5 m above ground level and central to each stable, providing an approximate light coverage of 4.5 x 4.5 x 3.5 m [49]. These lights provided blue-enriched white polychromatic light by day (peak wavelength 460 nm and mean light intensity 487.8 ± 41.5 lux) and dim red light (peak wavelength 620 nm and mean light intensity 9.7 ± 0.2 lux) at night. Lights transitioned gradually over 20 min from red to white at dawn (07:20) and white to red at dusk (20:20) to give approximately 13 h of daytime light when in use (Fig 2). Control lighting consisted of a 5 ft T8 fluorescent 58W 3000K tube light (Bell, Normanton, West Yorkshire, UK) secured 3.8 m above ground level and placed centrally in each stable, providing an approximate light coverage of 4 x 5 x 3 m, (peak wavelength 560nm and mean light intensity of 227.8 ± 7.5 lux). When in use, these were manually turned on each morning between 07:00 and 07:30 (no transition) and turned off each night between 20:00 and 21:00 (no transition), providing

| Group A | | | Group B | | |
|---|---|---|---|---|---|
| Horse 5 | Horse 4 | Horse 3 | Horse 10 | Horse 9 | Horse 8 |
| 10 year old | 10 year old | 14 year old | 17 year old | 8 year old | 12 year old |
| Gelding | Gelding | Gelding | Gelding | Mare | Gelding |
| Central aisle | | | | | |
| No horse | Horse 1 | Horse 2 | Horse 6 | No horse | Horse 7 |
| | 15 year old | 15 year old | 7 year old | | 15 year old |
| | Gelding | Mare | Gelding | | Mare |

**Fig 1. Schematic of the experimental barn layout and horse assignment to individual stables and groups.**

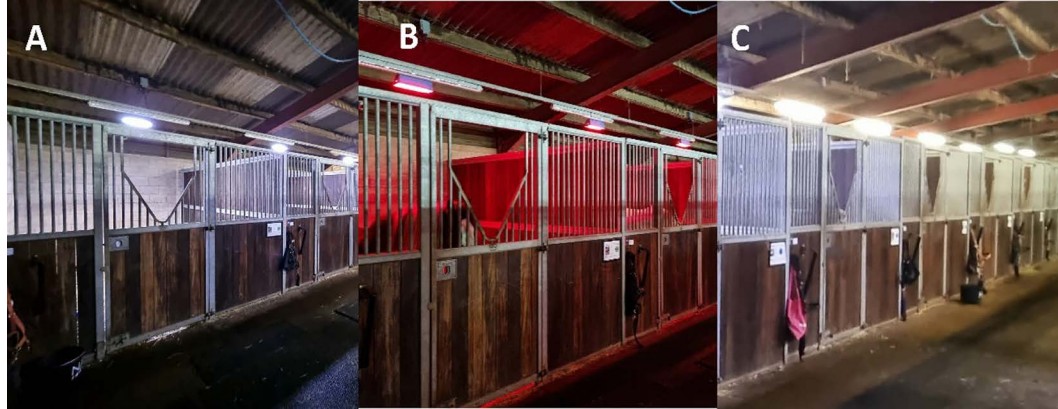

**Fig 2. Images of the stable lighting. (A)** Treatment nighttime (Equilume® Stable Light), **(B)** Treatment daytime (Equilume® Stable Light), and **(C)** Control daytime (fluorescent strip light).

approximately 13–14 h of daytime light and darkness conditions at night. Spectral light profiles from measurements taken at horse eye level using a CRI Illuminance Meter CL-70F are presented in Fig 3 and light intensity readings within each stable under each lighting condition (Treatment and Control) are detailed in Table 1.

Treatment and Control lighting systems were set up on different electrical circuits such that a single lighting system could be active within an individual stable at a time to facilitate the different experimental phases of the study. The stables did not have floor-to-ceiling partitions such that a slight spillover of light was visible in stables either side of the Group A/ Group B. However, this spillover did not impact the light intensities recorded for Day or Night within lighting condition as the intensity values recorded in stables either side of the treatment divide fell within the range for that group and did not vary from the group mean (Table 1).

### Lighting treatment periods

Horses experienced each lighting condition once over two four-week experimental periods. The first experimental period took place between January 23rd and February 18th, 2023, during which time Group A experienced the treatment lighting condition and Group B experienced the control lighting condition. The second four-week period took place between February 19th and March 19th 2023, when the lighting conditions for each group were switched. On week four of each experimental period, video footage of each horse in its stable was recorded continuously on days 1–3 for 72 h, hair sample collections were carried out on Days 4–5 at four-hour intervals starting at 08:00 for 52 h, and SBR recording occurred within a two-hour window from 17:00–19:00 on Days 1 and 3 for all horses (Table 2).

### Behavioural data collection

**Spontaneous blink rate.** Spontaneous blink rate data was measured by securely attaching Go-Pro Hero 10 cameras (7 x 5 x 4 cm Go-Pro Incorporated) to the headcollars of each horse individually using Same Top™ backpack mount attachments (8.5 x 5 x 4.3 cm, model number SA-YG16). Cameras were set up to record footage at a quality of 4k/60FPS at 1x speed. The camera was pointed at the left eye avoiding the animals' direct line of vision or contact with the eye [47]. Where necessary, horse's forelocks were plaited to avoid obstruction to the lens of the camera. Measurements were taken between 17:00 and 19:00, a normally quiet period for the yard and prior to the final evening forage ration arriving. Five cameras were used to simultaneously record SBR from Group A over 30 min, and then swapped to Group B horses for 30 min. A period of ten minutes was allowed to enable horses to acclimatise to the equipment and the presence of

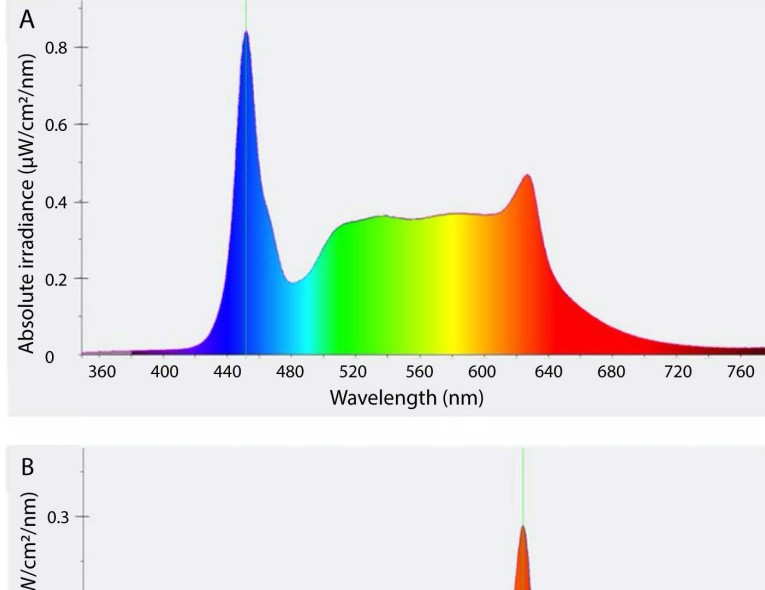

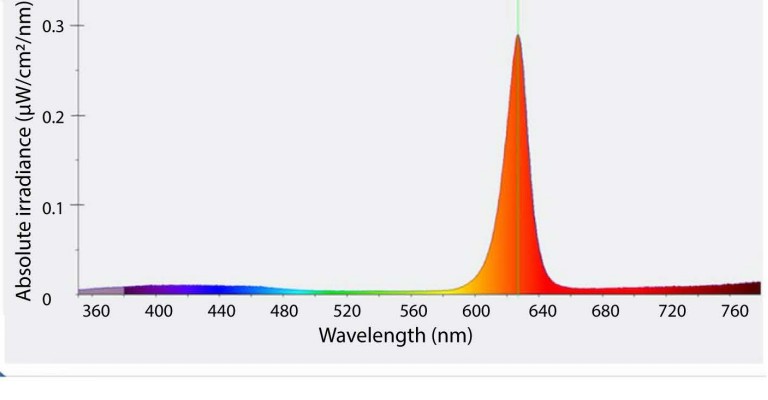

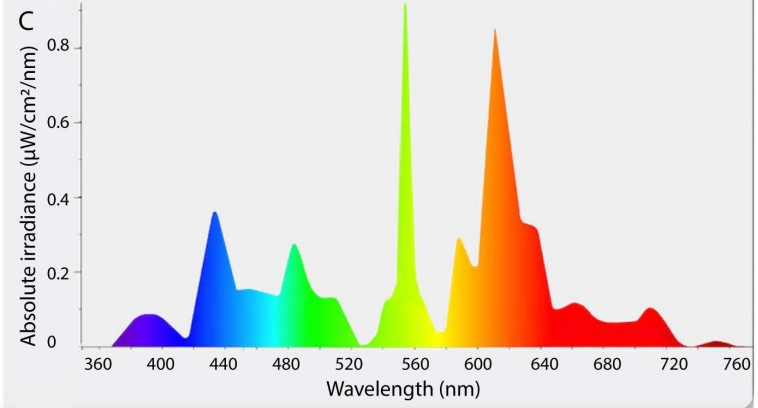

**Fig 3. Spectral light profiles. (A)** Blue-enriched white polychromatic light by day, **(B)** red light by night and **(C)** fluorescent light. Light measurements were taken using a CRI Illuminance Meter CL-70F at horse-eye level, with the light sensor facing upwards.

the researchers prior to recording. Recordings occurred twice for each horse on non-consecutive days to facilitate the calculation of a mean value for each individual and eye blink behaviour measured [50]. Horses were loosely tied via a lead-rope attached to a headcollar to a tie ring at the front of the stable for the duration of the recording and observed from a distance to ensure that cameras remained in position. Finally, full eye blinks, defined as momentary but full closure

**Table 1. Light intensity measurements (lux) captured during Day and Night from each stable using a CRI Illuminance Meter CL-70F when positioned centrally with the light sensor facing upwards at horse eye level height.**

| Animal | Treatment | | Control | |
|---|---|---|---|---|
| | Day (lux) | Night (lux) | Day (lux) | Night (lux) |
| Horse 1 | 368 | 10 | 234 | <1 |
| Horse 2 | 484 | 9.25 | 213 | <1 |
| Horse 3 | 575 | 11 | 212.5 | <1 |
| Horse 4 | 529.5 | 10.2 | 243 | <1 |
| Horse 5 | 552.5 | 8.6 | 266 | <1 |
| Horse 6 | 363.5 | 9.8 | 230.5 | <1 |
| Horse 7 | 401.5 | 10 | 248.25 | <1 |
| Horse 8 | 788 | 9.3 | 242.5 | <1 |
| Horse 9 | 430.5 | 10.1 | 194 | <1 |
| Horse 10 | 385 | 9 | 194.5 | <1 |
| Mean±SEM | 487.8±41.5 | 9.7±0.2 | 227.8±7.5 | <1 |

**Table 2. The data collection schedule for video recording, hair sample collection and spontaneous blink rate (SBR) recording during week four of each experimental period.**

| Video Recording | | | Hair Follicle Collection | | |
|---|---|---|---|---|---|
| 1 | 2 | 3 | 4 | 5 | 6 |
| Monday | Tuesday | Wednesday | Thursday | Friday | Saturday |
| Video Recording commenced at 0:00 for 72 hours. SBR recording between 17:00 and 19:00 | Video Recording. | Video recording concluded at 0:00. SBR recording between 17:00 and 19:00 | Hair follicle collection commenced at 08:00 at 4-hour intervals for 52 hours. | Hair follicle collection. | Hair follicle collection concluded at 08:00. |
| 0:00 | 0:00 | 0:00 | 0:00  04:00  08:00  12:00  16:00  20:00 | 0:00  4:00  8:00  12:00  16:00  20:00 | 0:00  04:00  08:00 |

of the eyelid [51] were quantified by a single trained observer using a mechanical tally counter (Tebery Products™) from the gathered video footage.

**Sleep behaviour.** Behavioural observations of horses within stables were recorded for consecutive 24-h periods over three consecutive days in each experimental condition (Table 2) using focal continuous sampling against a predefined ethogram (Table 3) and Hikivision CCTV equipment comprising a 16 channel 4K Power over Ethernet (PoE) Network Video Recorder (NVR) (Fig 4A) and 4MP Acusense fixed lens (2.8/4 mm) cameras (one per stable, Fig 4B) with 30 m infrared range. Cameras were installed in stables via network cables secured at a height that prevented horses from interfering with them. Footage was downloaded from all cameras at the end of the recording period onto a Seagate Portable 2TB, External Hard Drive. All files were then uploaded to a secure SharePoint site on MS OneDrive.

## Hair follicle collection for clock gene expression

Mane hair samples, complete with follicles, were collected from each horse at 4 h intervals for 52 h from 08:00 on day four of week four in each experimental period. Each sample comprised 10–20 hair which were carefully trimmed and placed in 2 ml tubes containing 0.75 ml of RNAlater® solution. Overnight sampling was conducted using red light headlights and torches. Samples were stored at ambient temperature (<18°C) while in transit for 48 h to the laboratory, conditions shown not to compromise RNA quality or quantity in hair follicle samples stored in RNAlater® [52]. On arrival, the samples were stored at 4°C for 72 h before being decanted and stored at −80°C until analysis.

**Table 3. Ethogram describing (sleep) behavioural states for horses (adapted from [5]).**

| Sleep State | Definition |
|---|---|
| Wakefulness (awake) | Horse is ingesting, drinking, excreting or moving, characterised by frequent ear movement and eyes open/ blinking. |
| Standing (awake) | Horse is immobile but there may be readjustment of limbs, itching, chewing licking etc. With or without resting hindlimb. The head is low/ close to vertical but poll will be above withers. Eyes open/half shut or frequent opening and shutting. Ear movement occurs but less frequently. |
| Standing (sleep) | After a distinct head drop where the poll is equal to or lower than the withers, the horse is standing immobile. Eye partially or completely shut and low frequency, if any, blinking. Little if any ear movement (ears not pointing forward). With or without resting hindlimb. |
| Sternal recumbency (sleep) | Horse is lying down, sternum in contact with ground, legs folded beneath. Head and limbs are immobile, eye partially or completely shut/ low frequency, if any, blinking. Little if any ear movement (ears not pointing forward). |
| Sternal recumbency (awake) | Horse is lying down, sternum in contact with ground, legs folded beneath. Large movements such as scratching or eating are observed as a state of wakefulness, further characterised by frequent ear movement and eyes open/blinking. |
| Lateral Recumbency (sleep) | Horse is lying on its side, legs extended, head on floor. Horse is largely immobile although there may be (limb/ body) twitches/ some ear movement. |
| Out of the stable (awake) | The time the horse leaves the stable to the time it returns. |
| Unknown | At any point during observation when the horses head moves out of view, i.e., we do not know if they are in a state of wakefulness of sleep. |

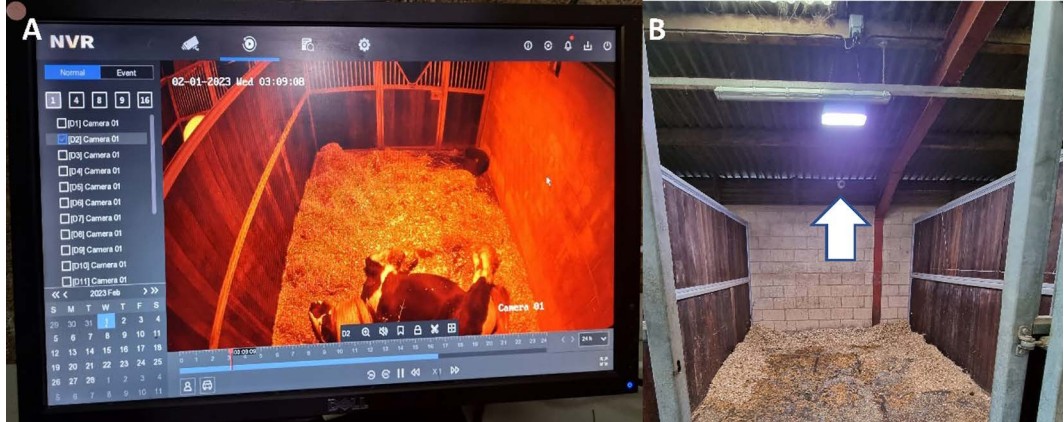

**Fig 4. Video recording within stables. (A)** Network Video Recorder visual of a stable and **(B)** placement of camera (denoted by white arrow) within the stable.

**RNA isolation.** Following removal from the freezer, the hair samples were further trimmed to just above the hair follicles and transferred to a nuclease-free microcentrifuge tube using forceps. RNA was isolated using the Quick-RNA Microprep Kit (Zymo Research, California, USA) as per the manufacturer's instructions with the following modifications: a single 5 mm steel bead was added to each tube before adding 500 µl of RNA Lysis Buffer. Following the addition of the Lysis Buffer the hair follicle samples were homogenized for two minutes at a frequency of 30 hertz using the Qiagen TissueLyser system. All remaining steps were as per the supplier's instructions except for the inclusion of a final two minute centrifugation step to ensure complete removal of any residual wash buffer. The RNA was eluted in 15 µl and stored at −80°C. RNA quantity was measured using the NanoDrop Spectrophotometer (Thermo Fisher Scientific, MA, USA) and RNA quality was assessed using the Agilent Bioanalyzer RNA 6000 (Agilent Technologies, CA, USA).

**Quantitative polymerase chain reaction (qPCR).** For each sample, 250 ng RNA was converted to complementary DNA (cDNA) in a 20 µl reaction using the Applied Biosystems High-Capacity cDNA Reverse Transcription Kit (Thermo Fisher) as per the manufacturer's instructions. A cDNA pool containing 3 µl from each sample was prepared and used to generate a 7-point, 1 in 4 serial dilutions for the standard curve, positive controls and inter-plate calibrators. Quantitative polymerase chain reaction (qPCR) was performed to detect the relative expression *NR1D2* (Nuclear receptor subfamily1, group D, member 2), *PER2* (Period homolog 2) and the clock-controlled gene *DBP* (D-site of albumin promoter binding protein) using the Applied Biosystems 7500 FAST Sequence Detection system (Thermo Fisher Scientific, MA, USA) and SYBR green chemistry (Roche, Burgess Hill, UK). These candidate genes were selected based on prior research indicating that they are expressed cyclically in human and horsehair follicles [23,24]. Primer design was performed using the Primer BLAST tool (https://www.ncbi.nlm.nih.gov/tools/primer-blast/) based on the equine genome sequence assembly available in the NCBI database https://www.ncbi.nlm.nih.gov/datasets/genome/GCF_002863925.1/, version EqCab3 GCA_002863925.1 All primer sequences are provided in S1 Table. The efficiencies of all primers used in the study were determined using the standard curve method and were shown to lie between 90% and 110%. The stability of eight potential reference genes was assessed for suitability as internal controls using the geNORM function within the qBASE+ (Ghent, Belgium) analysis software. *PPIA* (Peptidylprolyl Isomerase A) and *RPL19* (ribosomal protein L19) were shown to be the most stably expressed, with a geNorm M value >0.2 indicating very high reference gene stability and were selected to normalise the gene expression data. The qPCR reactions were prepared in duplicate in a 20µl reaction as per the manufacturer's instructions (FastStart Universal SYBR Green Master, Roche) using a final concentration of 300 nM for each primer. The thermal cycling involved one cycle at 50°C for two minutes, followed by one cycle at 95°C for 10 minutes and 40 cycles at 95°C for 15 sec, ending with 60°C for one minute.

## Data analysis

**Spontaneous blink rate.** Analysis via Shapiro-Wilk's test indicated that the data sets were normally distributed. Paired t-tests were used to compare mean values of full blinks under both lighting systems (GraphPad Prism version 10.3.1 for Windows [GraphPad Software, Boston, Massachusetts USA, www.graphpad.com]). Data was presented as means ± SEM and $P < 0.05$ was considered significant.

**Sleep behaviour.** In total, 1,440 hours of footage were recorded. To facilitate analysis (one hour of footage = 10–20 minutes to review), ten observers were trained on use of the ethogram and data recording using an Excel spreadsheet by the lead author. This comprised three meetings to 1) explain and refine the ethogram, 2) instruct on correct recording of observations from sample videos using a pre-prepared template spreadsheet, and 3) to clarify any remaining issues following completion of recording behaviours from sample videos used for completion of inter-observer reliability testing. Subsequently, each observer was assigned all the footage for one horse. Before data analysis, the lead author checked that the data for each horse on each of the recorded days totalled 24 h (+/- 60 sec) and reviewed the video footage to make corrections where necessary.

Intra-class correlation coefficient (ICC) tests conducted to determine the inter-observer reliability amongst the 10 observers prior to reviewing the experimental footage showed a high-reliability rating of 0.92 (ICC absolute agreement for single measures F = 88.6; P < 0.01). Results of inter-observer reliability tests supported the consistency of the observed behaviour reporting from the ten observers.

Total sleep duration was calculated as the summation of 'lateral recumbency (sleep)', 'sternal recumbency (sleep)' and 'standing (sleep)' data combined; total recumbency duration was calculated by the summation of 'lateral recumbency (sleep)', 'sternal recumbency (sleep)' and 'sternal recumbency (awake)' data combined. In addition, 'sternal recumbency (sleep and awake)', 'lateral recumbency (sleep)', 'wakefulness', 'standing (awake)', 'out of-stable' and 'unknown' behaviour durations were analysed individually. All behaviour data underwent statistical testing using GraphPad Prism version 10.3.1 for Windows. All data were first assessed for normality and lognormality using D'Agostino & Pearson tests.

Statistical differences between lighting conditions (Treatment versus Control) and between lighting periods ('Day versus 'Night') were assessed using paired t-tests or Wilcoxon tests where data was shown to be normally or not normally distributed, respectively. The lighting period 'Day' was defined as behaviours that occurred between 07:20–20:20 during the experimental observation periods. The lighting period 'Night' was defined as behaviours that occurred between 20:21 and 07:19 during the experimental observation periods.

One-way repeated measures ANOVA or a Friedman's test was conducted to determine the impact of the lighting period (Day or Night) on the horses' behaviour depending on data normality followed by Tukey's or Dunn's multiple comparisons tests, where appropriate, to examine differences between and within each lighting condition during each lighting period. These tests assessed differences across the four groups: Control Night, Control Day, Treatment Night and Treatment Day. Data were considered statistically significant if P < 0.05.

**Hair follicle clock gene expression.**  Data from hair follicle samples for all 10 horses under each lighting condition were assessed. Calibrated Normalized Relative Quantities (CNRQ) were calculated for each sample and each gene transcript using the qBASE+ software package (Ghent, Belgium) and the two reference genes RPL19 and PPIA. The data were normalised within horse to account for individual clock gene expression abundance differences. Repeated measures two-way ANOVA was used to assess the effect of time, treatment, and time x treatment interaction on gene expression in all horses (n = 10) (GraphPad Prism version 10.3.1 for Windows). Data were considered statistically significant if P < 0.05 and were presented as means ± S.E.M. Where repeated measures two-way ANOVA found significant differences in the effect of time, cosinor analysis [53] based on the least squares cosine fit method [54] was used to assess the presence of 24 h (circadian) periodic variance. This cosinor method produced an estimate of acrophase (time of peak value) and robustness (stability).

## Results

### Spontaneous blink rate

No differences were detected for full blink rate between control (290 ± 18.32) and treatment (264.7 ± 34.61) lighting conditions (P > 0.05; Fig 5B).

### Sleep behavioural data

No differences were observed in the mean duration of each behavioural parameter exhibited between Treatment or Control lighting conditions (P > 0.05). Day versus Night differences existed for each behavioural parameter regardless of lighting condition (P < 0.05) with the exception of Standing (awake) and Unknown (P > 0.05) (Table 4).

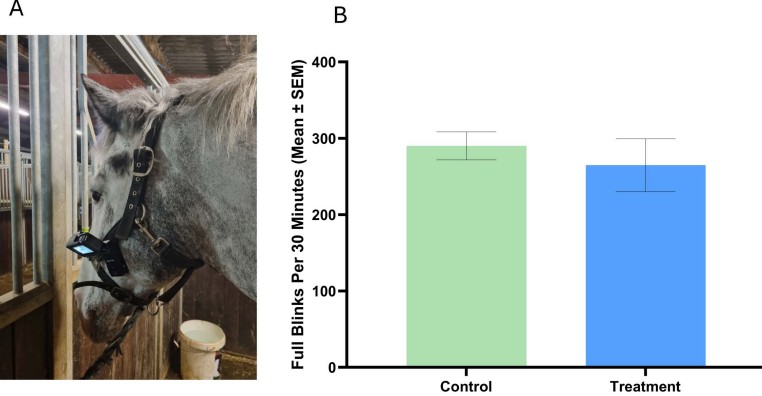

**Fig 5.  Spontaneous blink rate. (A)** Example of Go-Pro Hero 10 camera (7 x 5 x 4 cm) attached to the headcollar of a horse for collection of spontaneous blink rate data. **(B)** Comparison of Full Blinks between Fluorescent (control) and LED (treatment) lighting conditions presented as means ± SEM.

**Table 4. Comparison of behaviour patterns between Control and Treatment lighting conditions and between Day and Night.**

| | LED Lighting Treatment | | | Lighting Period | | |
|---|---|---|---|---|---|---|
| | Control (n = 10) | Treatment (n = 10) | P-value | Day (n = 10) | Night (n = 10) | P-value |
| Behavioural parameter | Duration in sec (duration in h) Mean ± SEM | | | Duration in sec (duration in h) Mean ± SEM | | |
| Total Sleep[a] | 10967 ± 2045 (3.04 ± 0.57) | 10871 ± 1908 (3.02 ± 0.53) | 0.945 | 1535 ± 428.9 (0.43 ± 0.12) | 9421 ± 1526 (2.62 ± 0.42) | <0.001 |
| Total Recumbency[b] | 9441 ± 1406 (2.62 ± 0.39) | 9245 ± 1357 (2.57 ± 0.38) | 0.770 | 200.9 ± 52.58 (0.06 ± 0.01) | 8713 ± 1320 (2.42 ± 0.37) | <0.001 |
| Sternal Recumbency[c] | 7343 ± 998.1 (2.04 ± 0.28) | 7268 ± 992.8 (2.02 ± 0.28) | 0.695 | 184.4 ± 49.45 (0.05 ± 0.01) | 6795 ± 939.8 (1.89 ± 0.26) | <0.001 |
| Lateral Recumbency | 2098 ± 711.6 (0.58 ± 0.2) | 1975 ± 651.9 (0.55 ± 0.18) | 0.765 | 16.45 ± 11.02 (0.005 ± 0.003) | 1918 ± 633.9 (0.53 ± 0.18) | 0.016 |
| Wakefulness | 44692 ± 2076 (12.41 ± 0.58) | 42949 ± 1344 (11.93 ± 0.37) | 0.248 | 24679 ± 1013 (6.86 ± 0.28) | 18191 ± 1118 (5.05 ± 0.31) | 0.004 |
| Standing (awake) | 12453 ± 2120 (3.46 ± 0.59) | 15433 ± 2288 (4.29 ± 0.64) | 0.105 | 7229 ± 1395 (2.01 ± 0.39h) | 7394 ± 850.5 (2.05 ± 0.24h) | 0.870 |
| Standing sleep | 4809 ± 1610 (1.34 ± 0.45) | 4415 ± 1235 (1.23 ± 0.43) | 0.786 | 1455 ± 421.6 (0.40 ± 0.12) | 3533 ± 887.9 (0.98 ± 0.27) | 0.005 |
| Out of stable | 6050 ± 491.8 (1.68 ± 0.14) | 6988 ± 407 (1.94 ± 0.11) | 0.152 | 6471 ± 403.7 (1.80 ± 0.11) | 0 ± 0 (0 ± 0) | N/A |
| Unknown | 7530 ± 2460 (2.09 ± 0.68) | 5885 ± 2199 (1.63 ± 0.61) | 0.469 | 2288 ± 889.7 (0.64 ± 0.24) | 4461 ± 1743 (1.24 ± 0.48) | 0.065 |

P-value of <0.05 was considered significant.

Sec: seconds; h: hours; SEM: standard error of means;

[a]Total Sleep includes the following behaviours: Sternal Recumbency (Sleep), Lateral Recumbency (Sleep) and Standing (Sleep).

[b]Total Recumbency includes the following behaviours: Sternal Recumbency (Sleep), Sternal Recumbency (Awake) and Lateral Recumbency (Sleep).

[c]Sternal Recumbency includes the following behaviours: Sternal Recumbency (Sleep), Sternal Recumbency (Awake).

[*]Not Applicable (N/A); The horses were not out of the stable during the night.

## Total sleep

Data for total sleep duration (comprising lateral recumbency, sternal recumbency sleep, and standing sleep) were found to be normally distributed. No overall effect of treatment conditions was observed (P = 0.945; Table 4) with a mean total sleep duration of 3.04 h for horses under the control lighting condition and 3.02 h under the treatment lighting condition. Data for the lighting period (Day and Night) were normally distributed. The lighting period affected total sleep duration, with higher values observed during the night (P < 0.001; Table 4). Multiple comparison tests revealed differences within each lighting condition for Day vs Night (P < 0.001: Table 5) but no differences were observed between Control Day and Treatment Day (P = 0.333) or Control Night and Treatment Night (P = 0.995). Horses displayed inter-individual variation in behavioural profiles for sleep (Fig 6).

## Total recumbency

Data for total recumbency (comprising lateral recumbency, sternal recumbency sleep and sternal recumbency awake) were found to be not normally distributed. No overall effect of treatment condition was observed (P = 0.77; Table 4; Fig 7A). Data for the lighting period (Day vs Night) were normally distributed. The lighting period had an effect on total recumbency duration, with higher values observed during the night (P < 0.001; Table 4). Multiple comparison tests (Table 5) revealed differences within each lighting condition for Day vs Night (P < 0.001) but no differences were observed between Control Day and Treatment Day (P > 0.999) or Control Night and Treatment Night (P > 0.999; Fig 7B and 7C).

**Table 5. Multiple comparisons analyses of the behaviour patterns total sleep, total recumbency and wakefulness for Control and Treatment conditions by the Lighting Period (Day and Night).**

**Total Sleep duration[a] (sec)**

|  | Mean | SEM | P-value | Multiple Comparisons P-value | | | |
|---|---|---|---|---|---|---|---|
|  |  |  |  | Control Day | Treatment Day | Control Night | Treatment Night |
| Control Day | 2065 | 421.3 | <0.001 | – | 0.333 | 0.003 | 0.003 |
| Treatment Day | 1205 | 490.3 |  | 0.333 | – | >0.001 | >0.001 |
| Control Night | 9299 | 1624 |  | 0.003 | <0.001 | – | 0.995 |
| Treatment Night | 9545 | 1596 |  | 0.003 | <0.001 | 0.995 | – |

**Total recumbency duration[b] (sec)**

|  | Mean | SEM | P-value | Multiple Comparisons P-value | | | |
|---|---|---|---|---|---|---|---|
|  |  |  |  | Control Day | Treatment Day | Control Night | Treatment Night |
| Control Day | 458.5 | 244.1 | <0.001 | – | >0.999 | 0.034 | 0.019 |
| Treatment Day | 18.15 | 17.87 |  | >0.9999 | – | 0.002 | >0.001 |
| Control Night | 5759 | 1109 |  | 0.0335 | 0.002 | – | >0.9999 |
| Treatment Night | 6023 | 987.8 |  | 0.0194 | 0.001 | >0.9999 | – |

**Wakefulness duration (sec)**

|  | Mean | SEM | P-value | Multiple Comparisons P-value | | | |
|---|---|---|---|---|---|---|---|
|  |  |  |  | Control Day | Treatment Day | Control Night | Treatment Night |
| Control Day | 23348 | 1995 | 0.007 | – | 0.998 | 0.314 | 0.044 |
| Treatment Day | 23736 | 1218 |  | 0.998 | – | 0.247 | 0.028 |
| Control Night | 19316 | 1640 |  | 0.314 | 0.247 | – | 0.605 |
| Treatment Night | 17051 | 1186 |  | 0.044 | 0.028 | 0.605 | – |

P-value of < 0.05 was considered significant.

Sec: seconds; SEM: standard error of means.

[a]Total Sleep includes the following behaviours: Sternal Recumbency (Sleep), Lateral Recumbency (Sleep) and Standing (Sleep).

[b]Total Recumbency includes the following behaviours: Sternal Recumbency (Sleep), Sternal Recumbency (Awake) and Lateral Recumbency (Sleep).

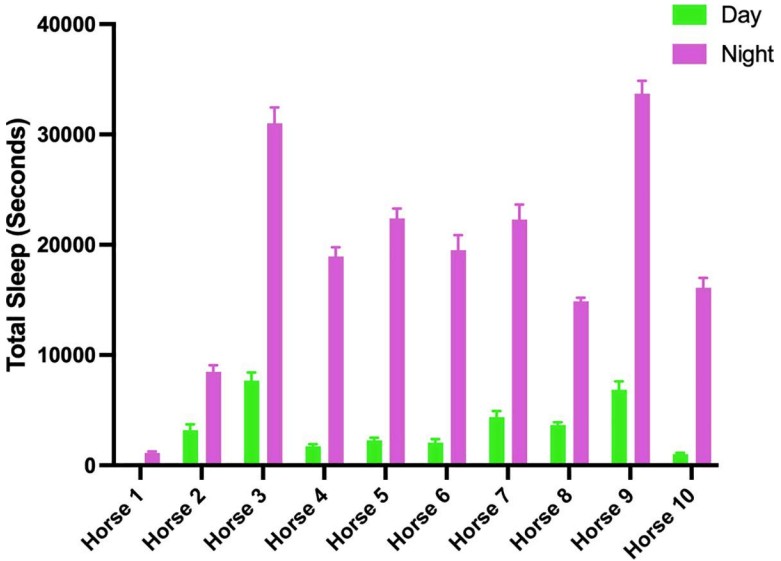

**Fig 6. Total sleep duration during Day and Night for three consecutive 24 h periods for each individual horse.** Data are presented as means±SEM.

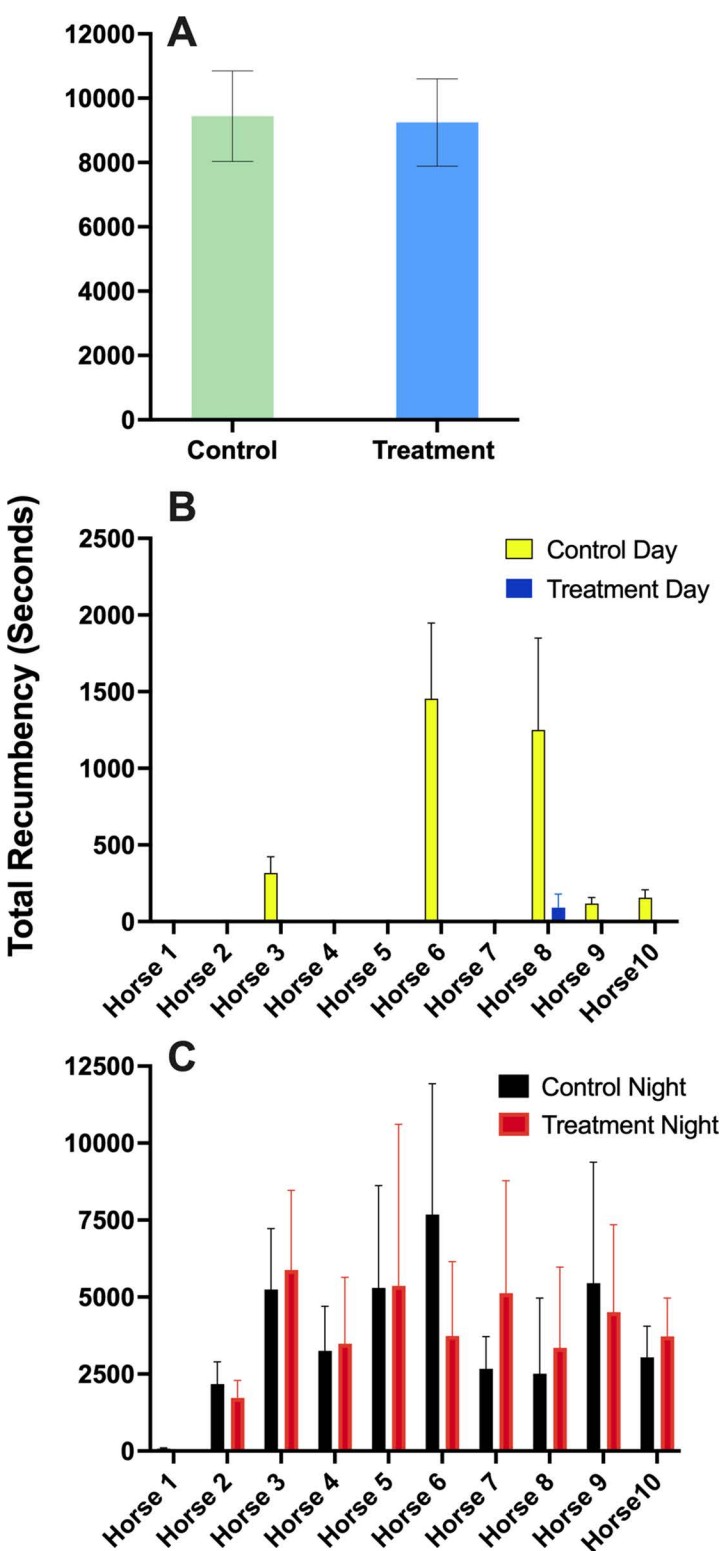

**Fig 7. Total recumbency duration of horses maintained under customised LED lighting (treatment) and fluorescent lighting (control) for three consecutive days.** (A) Total recumbency duration during Day (B) and Night (C) for each lighting condition (control and treatment). Data are presented as means ±SEM for three consecutive 24-h periods.

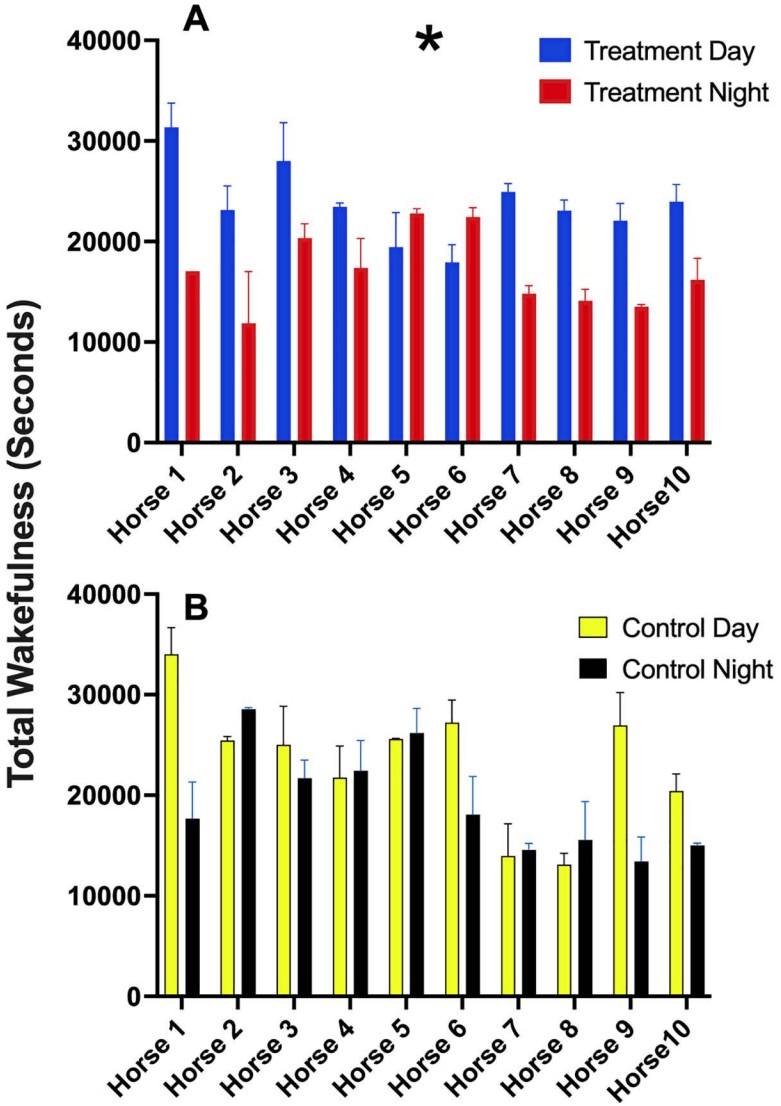

**Fig 8. Day versus night durations in wakefulness for three consecutive days. (A)** Day versus night duration under treatment (LED) and **B)** Day versus night duration under control (fluorescent) lighting conditions Data are presented as means+/-SEM for three consecutive 24 h periods. * indicates P < 0.05.

## Sternal and lateral recumbency

The data for sternal recumbency were found to be not normally distributed. No overall effect of treatment condition was observed (P = 0.695). The data for lateral recumbency were found to be not normally distributed. No overall effect of treatment condition was observed (P = 0.765). The lighting period had an effect on both sternal (P < 0.001) and lateral recumbency (P = 0.016) duration, with higher values observed during the night (Table 4). Multiple comparison tests revealed differences within each lighting condition for Day vs Night (P < 0.001) but again no differences were observed for either sternal or lateral recumbency durations between Control Day and Treatment Day or Control Night and Treatment Night (P > 0.999 for all, Table 5).

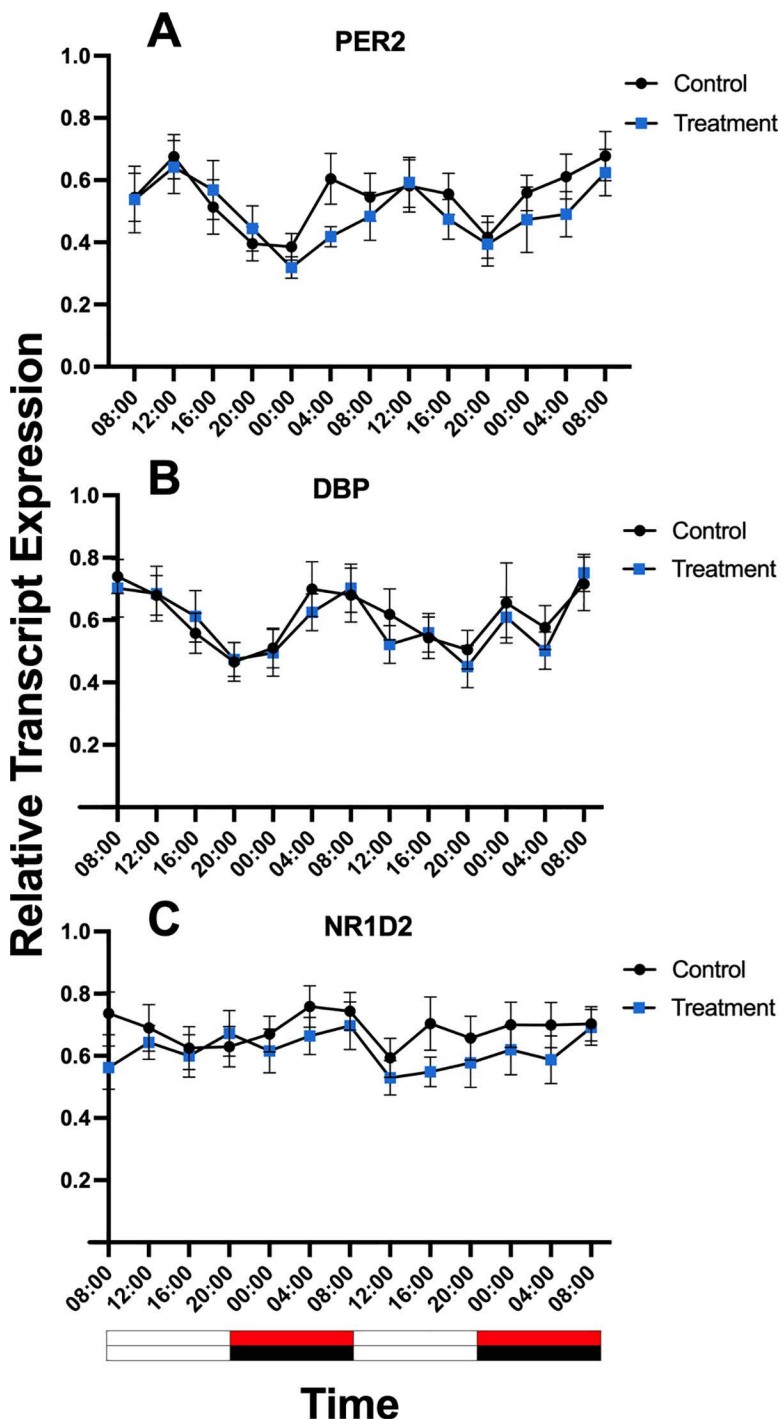

**Fig 9. Profiles of clock gene expression in hair follicles from horses (n = 10) maintained under customised LED lighting (treatment) or fluorescent (control) lighting conditions. (A)** *PER2*, **(B)** *DBP* and **C)** *NR1D2*. Data are presented as means ± SEM. There was a significant overall effect of time for *PER2* and *DBP* (P < 0.01; P < 0.001), but no effect of treatment or time by treatment interaction (P > 0.05). No effect of time, treatment or time by treatment interaction was detected for *NR1D2* (P > 0.05). White and red or black bars beneath the graphs represent periods of daytime light and red light or darkness at night, respectively.

## Wakefulness

Data for wakefulness were found to be not normally distributed. No overall effect of treatment condition was observed (P=0.248; Table 4). Data for lighting period (Day and Night) were normally distributed. The lighting period had an effect on wakefulness duration, with higher values observed during the day (P<0.004; Table 4). Multiple comparison tests revealed Day vs Night differences (P=0.007), with higher wakefulness duration by day for the treatment condition only (P=0.028). No differences were observed for the control lighting condition for Day vs Night (P=0.314; Table 5). No differences were observed between Control Day and Treatment Day (P=0.998) or Control Night and Treatment Night (P=0.605). Data for each horse during each lighting condition and lighting period are presented in Fig 8.

## Additional behaviour parameters

There were no differences in mean durations between lighting condition for the parameters standing (awake) (P=0.105), standing sleep (P=0.786), out of stable (P=0.152) and unknown (P=0.469; Table 4). No differences were observed between Day and Night for standing awake (P=0.87) and unknown (P=0.065), and higher values for standing sleep were observed during Night (P=0.005; Table 5).

## Clock gene expression

Expression of the core clock genes *PER2, NR1D2*, and the clock-controlled gene *DBP* were detected in all samples (Fig 9). The data was normally distributed. There was an overall effect of time for *PER2* and *DBP* (P<0.01; P<0.001), but no effect of treatment or time by treatment interaction (P>0.05). No effect of time, treatment or time by treatment interaction was detected for *NR1D2* (P>0.05).

**Cosinor analysis results.** Cosinor analysis detected 24-h rhythmicity for both *PER2* and *DBP* (P<0.01) in control and treatment lighting conditions (Table 6).

## Discussion

Appropriate exposure to lighting that mimics the environmental 24 h cycle is considered a key factor in the correct functioning of the mammalian circadian system, which in turn confers improved rest/activity rhythms and ultimately benefits well-being. This study investigated the influence of a customised LED lighting system and standard fluorescent lighting fixtures on arousal, sleep behaviours and circadian rhythmicity in stabled horses for the first time. The results reveal no overall difference in mean durations of any parameter over 24-h measured in horses maintained under the two different lighting conditions. Under both conditions, sleep and wakefulness durations were similar and strong circadian rhythmicity was evident.

### Effects of lighting on behaviour measures

The lack of a treatment effect on spontaneous blink rates could be attributed to the management conditions of the studied group, which provided visual contact with conspecifics, ample forage, daily exercise and sufficient environmental stimulation with regular human interaction. In this low-stress context, the stress coping benefits afforded by the provision

**Table 6. Cosinor analysis of 24-h clock gene expression profiles in hair follicles. Robustness (rhythm stability), acrophase (time of rhythm peak) and P-values are presented for horses maintained in treatment and control lighting conditions.**

| Gene Transcript | Robustness % | | Acrophase (24-h) | | P-Value | |
|---|---|---|---|---|---|---|
| | Treatment | Control | Treatment | Control | Treatment | Control |
| *PER2* | 68.8 | 57.5 | 11:24 | 08:58 | 0.00 | 0.01 |
| *DBP* | 74.1 | 57.0 | 08:53 | 07:47 | 0.01 | 0.00 |

of optimised light spectra and intensities may not have been sufficient to induce blink rate changes within the studied population. In support of this notion, horse based studies that utilised SBR as a stress indicator applied clearly defined acute stressors such as sham clipping [48] and sudden arrival of a novel object [47]. As such, future studies featuring higher stress management conditions are required to explore further the relationship between lighting conditions and stress.

The horses in the study achieved an average total sleep time of three hours, but with substantial variation between individuals. These results align with findings of previous studies in stabled horses where a range of total sleep time from 2.5 to 5 h was reported [8]. No difference in total sleep time was observed between treatment and control groups, but as expected, sleep duration significantly differed between lighting periods (Day and Night). In agreement a with previous study, it was observed that horses carried out most sleep behaviour during the night [55]. As diurnally active prey animals, horses may feel less safe and exhibit greater vigilance during daylight hours when near people and other external stimuli [56]. Greater wakefulness exhibited by day was therefore expected, but significant differences in wakefulness duration between day and night were only observed under the treatment condition. One explanation for this relates to the known increased suppressive effect of blue-enriched lighting on melatonin production [35]. This is supported by the substantially lower levels of blue wavelengths in the spectral composition of the fluorescent lights used in the control condition, and the higher daytime light intensities recorded at eye level under the treatment LED lighting. Blue light is known for enhancing wakefulness in humans [57–60] and is also known to improve alertness in the workplace [61]. Our results suggest that the differences in proportion of blue light emitted by the two lighting conditions may have influenced alertness during daytime.

Horses were observed to spend more time standing during the day than at night. This finding is not unexpected as horses tend to exhibit more sleeping behaviour during the night [55]. It is common among horses to spend significant time standing during the day [62]. In feral horses, standing behaviour was described as a resting state characterised by a sense of ease and relaxation [63]. Standing can be seen as a resting behaviour as horses adopt the stay apparatus position as they stand at rest [64], allowing them to do so with minimal muscular effort. In this study, observers employed to study video files were found to significantly agree in their scores, however further clarification is required to help differentiate between standing awake, standing at rest (drowsiness) and standing sleep to increase the accuracy of reporting within the equine sleep literature [65]. While a distinct head drop is often indicative of the transition between drowsiness and sleep while standing [64], this was not always clearly perceived in all horses, requiring increased reliance on observation of partially or fully closed eyes and reduced blinks and ear movements to indicate sleep.

Unusually, Horse 1 did not display any recumbent behaviour during the six days of video observation contributing to a low total sleep duration. Although we do not know if the horse displayed recumbency behaviour on non-recorded nights of the study, it could be a welfare concern as lack of recumbency suggests REM sleep deprivation [66]. However, frequent daily horse welfare checks did not note any abnormalities for this animal. The observer also reported that this horse displayed frequent atypical behaviour at night, involving short periods during standing where the horse became unsteady, its head dropped, and its muzzle touched the ground. Such behaviour has been described for horses that do not lie down [64] and requires further investigation. Important to note was that the mean duration of 'Unknown' behaviour for Horse 1, where his head was over the stable door, was considerably longer at 6.5 h, compared to the mean 'Unknown' duration for all horses of 1.8 h. This difference implies that a significant proportion of sleep time for this horse may have occurred while his head rested on the stable door.

The lack of difference in duration of sleep behaviours between treatment and control lighting conditions implies that dim red light at night does not impact the normal sleep patterns in horses compared to natural dark conditions. This is the first study to confirm that red light is a suitable alternative to darkness in stabled horses to maintain sleep behaviour. These results support a previous finding that the nighttime rise in melatonin was maintained under dim red light [39] and further support using red light at night to avoid the disruptive effects of white light on circadian rhythms [38].

## Effects of lighting on circadian oscillations in hair follicles

Hair follicles represent a peripheral clock in mammals that is synchronised by the master clock and entrained by the light-dark cycle [67]. The expression patterns of the clock genes, *PER2, NR1D2,* and the clock-controlled gene *DBP,* have been used previously to assess internal circadian synchrony in human and horse studies [23,24,36]. In this study, significant circadian oscillations of *PER2* and *DBP* were observed under both lighting conditions indicating that both provided sufficient entrainment cues to the circadian system. Unexpectedly, *NR1D2* did not vary over time in contrast to previous findings. The lack of rhythmicity for this clock gene transcript may be due to the different stage of the hair growth cycle of the study horses as a result of exposure to a different photoperiod length compared to previous studies. In Collery et al., (2023), for example, horses were exposed to a long day photoperiod consisting of 17 h of light, contrasting with the photoperiod length of 13 h of light in the current study [36]. The duration of melatonin secretion, which mirrors the hours of darkness within the photoperiod, regulates the circannual pattern of prolactin secretion, and in turn the hair growth cycle [68–70]. It is postulated therefore that the lack of rhythmicity in NR1D2 may relate to the reported different roles played by specific clock genes in the regulation of the hair cycle at different growth stages in response to differing photoperiods [71].

Previously, stabled racehorses maintained under control conditions consisting of erratic nighttime exposure to incandescent light did not demonstrate rhythmic clock gene expression in hair follicles [36]. However, when switched to a customised LED lighting system, similar to that used in the current study, significant circadian clock gene rhythmicity for *NR1D2* and *PER2* was reported [36]. The authors concluded that the treatment lighting provided a robust entrainment signal to the circadian system, noting that red light at night enabled handlers to perform their duties during hours of darkness while avoiding the suppressive effects on melatonin production that switching on of white lights is known to induce. In contrast, the exposure to uninterrupted darkness experienced under the control lighting condition in the current study, which is known to facilitate the normal pattern of melatonin production, likely explains the differences in results in clock gene expression patterns between studies.

Many horse owners conserve energy by turning off barn lights during the day when not interacting with their horses. In modern equine husbandry, horses are often stabled for up to 23 h a day [72,73] and in the current study horses were stabled for at least 22 h. However, the experimental conditions in the current study were such that all horses experienced uninterrupted exposure to artificial light from morning until evening. This consistent exposure to daytime lighting likely contributed to the observed rhythmic clock gene expression patterns in all study horses. Further exploration of circadian rhythm and behaviour patterns of horses exposed to lighting programmes that vary in consistency and spectral composition would permit the development of best practice guidelines for stabled horses.

## Limiting factors

Due to the objectives of this study, such that the impacts of lighting were assessed in horses housed in their usual stables on a working yard, it was not feasible to block natural light from barn doorways. Daylight is naturally high in blue light, and due to the positioning of the stables within the barn, it is likely that some of this reached the study horses during the trial periods, especially when they hung their heads over the stable doors. This may have compromised the control lighting condition to some extent by increasing the blue light exposure during the day. More insights may be gained in future controlled studies where stables lack exposure to natural light.

Finally, the large inter-individual differences observed between horses in relation to their sleep behaviour patterns, especially Horse 1, may have limited the ability to detect differences in some cases, despite the implementation of a cross-over study design. Pre-screening of study animals to select for horses that clearly exhibit sleep durations within a range considered normal for the species may represent a valid consideration for future similar studies.

## Conclusion

No overall significant differences in arousal, sleep behaviour or clock gene rhythmicity in a peripheral tissue were observed in horses exposed to fluorescent daytime light and darkness at night or a customised LED lighting system comprising blue-enriched daytime light and red light at night. The horses in this study were well managed regarding adherence to turning lights on and off at regular times in the morning and evening. Therefore, the lack of significant findings is not unexpected given the lack of nighttime light disturbances under the control lighting condition. The findings also suggest that dim red light at night is a suitable light spectrum to facilitate normal sleep behaviour in horses as well as maintaining internal circadian synchrony. These results contribute to our understanding of how stable lighting can promote healthy sleep behaviour patterns and optimise circadian rhythms in horses, providing a foundation to inform future research in this important area.

## Supporting information

**S1 Table. Equine candidate clock gene and internal reference gene sequences used for qPCR.**
(DOCX)

## Acknowledgments

With thanks to Eleanor Taylor, Lorna Cameron, Megan Long, and Kate Baldwin for the extensive time dedicated to observing video files of horses within this study.

## Author contributions

**Conceptualization:** Linda Greening, Jane M. Williams, Barbara A. Murphy.

**Data curation:** Eilis Harkin.

**Formal analysis:** Linda Greening, Eilis Harkin, Panoraia Kyriazopoulou, John A. Browne, Barbara A. Murphy.

**Funding acquisition:** Linda Greening, Jane M. Williams, Barbara A. Murphy.

**Investigation:** Linda Greening, Eilis Harkin, Zoe Heppelthwaite, Francesca Aragona, Andrew Hemmings, Barbara A. Murphy.

**Methodology:** Panoraia Kyriazopoulou, Zoe Heppelthwaite, Francesca Aragona, John A. Browne, Jane M. Williams, Barbara A. Murphy.

**Project administration:** Jane M. Williams.

**Resources:** Andrew Hemmings, Barbara A. Murphy.

**Supervision:** Linda Greening, John A. Browne, Andrew Hemmings, Barbara A. Murphy.

**Validation:** Linda Greening.

**Visualization:** Panoraia Kyriazopoulou.

**Writing – original draft:** Linda Greening, Eilis Harkin, Barbara A. Murphy.

**Writing – review & editing:** Linda Greening, Eilis Harkin, Panoraia Kyriazopoulou, Zoe Heppelthwaite, Francesca Aragona, John A. Browne, Andrew Hemmings, Jane M. Williams, Barbara A. Murphy.

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
