## [Decision Letter · Decision Letter 0]

PONE-D-24-58876Influence of lighting on sleep behaviour, circadian rhythm and spontaneous blink rate in stabled riding school horses (Equus caballus)PLOS ONE

Dear Dr. Murphy,

Thank you for submitting your manuscript to PLOS ONE. After careful consideration, we feel that it has merit but does not fully meet PLOS ONE’s publication criteria as it currently stands. Therefore, we invite you to submit a revised version of the manuscript that addresses the points raised during the review process.

The requested changes are both in style and in substance. Specifically, the manuscript is both verbose and repetitive. The manuscript is long and could be significantly shortened without diminishing its impact by correcting these issues and by removing some items, as suggested by the reviewers. Furthermore, the link between SBR and sleep deprivation, as it pertains to horses, is circumstantial and the authours should be cautious with the interpretations of these data.

We look forward to receiving your revised manuscript.

Kind regards,

Paul A. Bartell

Academic Editor

PLOS ONE

2. Please amend your list of authors on the manuscript to ensure that each author is linked to an affiliation. Authors’ affiliations should reflect the institution where the work was done (if authors moved subsequently, you can also list the new affiliation stating “current affiliation:….” as necessary).

3. Please provide captions for Figure 9 in your manuscript.

[I have read the journal's policy and the authors of this manuscript have the following competing interests: BAM is the Founder of Equilume Ltd., a spin-out company deriving from her research program as associate professor at University College Dublin and is a member of the company’s Board of Directors. BAM is a shareholder in Equilume Ltd. PK is an employee of Equilume Ltd. The treatment lighting condition in the present study comprised an Equilume Stable Light and is a commercially available product.].

5. Please provide a complete Data Availability Statement in the submission form, ensuring you include all necessary access information or a reason for why you are unable to make your data freely accessible. If your research concerns only data provided within your submission, please write "All data are in the manuscript and/or supporting information files" as your Data Availability Statement.

Reviewers' comments:

Reviewer's Responses to Questions

**Comments to the Author**

1. Is the manuscript technically sound, and do the data support the conclusions?

Reviewer #1: Yes

Reviewer #2: Partly

2. Has the statistical analysis been performed appropriately and rigorously? 

Reviewer #1: Yes

Reviewer #2: I Don't Know

3. Have the authors made all data underlying the findings in their manuscript fully available?

Reviewer #1: Yes

Reviewer #2: Yes

4. Is the manuscript presented in an intelligible fashion and written in standard English?

Reviewer #1: Yes

Reviewer #2: Yes

5. Review Comments to the Author

Reviewer #1: Overall comments: An interesting study. I enjoyed reading this manuscript. There are some edits to be made, and those are noted below.

Specific comments:

Line 50-51: please recheck grammar for this sentence, it seems choppy jumping from talking about environmental factors in horses and then light stimuli. Perhaps it needs to be 2 separate sentences?

Line 57-60: need an additional comma after hypothalamus

Line 76-77: tie back in the connect to melatonin in the final sentence of this paragraph.

Line 78-81: what about melatonin? The light type that stimulates the circadian clock is discussed, but the way the sentence structure is set up, it feels like it is currently lacking a mention of the optimal stimulation for melatonin. OR – rephrase the sentence. Perhaps the part after the semicolon should just be its own sentence.

Line 97-101: is there research in any other mammalian species on red light and its impact on melatonin and the circadian rhythm?

Line 131-133: please provide number of mares and geldings, as well as breed.

Line 142-143: what type of forage? Be more specific

Line 135: do you mean the stalls were partitioned by wooden panels? Not the stables?

Line 152: Table 1 – just to clarify – the treatments were two separate barns? Or the same barn? It is unclear from this picture. If it is two separate barns, I would label the table to clarify.

Line 187 and line 188: barn and stable are used here interchangeably. Please pick one term to use and correct throughout paper.

Line 194-195: So there was no washout period in between the two trials? Is there literature to show this has no effect on variables being measured?

Line 200: I don’t think table 3 is really necessary. The protocol is explained sufficiently in the prior paragraph.

Line 347: consider moving the text in row 3 ( duration of behaviors in sec) to the bottom of the table – it’s a little jarring where it is now. And why for the results in this table are then H shown in all the rows, but not sec? I would either include both or neither, since you explain the units in row 3.

Line 438: was there any effect of age on the horses on behavior and arousal?

Line 489-498: Frankly, I do not this paragraph is necessary. There was no significant effect found by the researchers on time spent outside, and this did not seem to be a primary point of interest for the study. Especially considering the horses did different things when outside. While I agree looking at exercise and sleep patterns would be interesting, given the length of this paper, and sheer speculative manner of this particular paragraph, I would cut it.

Figure 2: are there better-quality images for this figure? The ones provided are a bit blurry.

Figure 3: were any pre-trials done to see if the location of the go-pro impacted horses lying down to sleep? It seems based on this picture, if the go-pro happened to be on the side the horse preferred to be on when lying down, that this could skew results.

Figure 6-10: again, the resolution of these images seems a little blurry. Would recheck to make sure the best quality is uploaded for the manuscript.

Reviewer #2: Review of “Influence of lighting on sleep behaviour, circadian rhythm and spontaneous blink rate in stabled riding school horses (Equus caballus)”.

Overview: This manuscript describes a study in which horses were exposed to 2 lighting sources and a host of behavioral and physiologic response variables were evaluated. The results indicate that in general there were not detectable differences between the treatment and control groups. There were clear differences in behavior between day and night and there was good evidence for circadian rhythms. This manuscript could be improved by focusing on the clear and obvious findings. It is likely that a manuscript that was 1/3 to ½ the length would be much more clear and concise and be of value to those that are conducting research in this area.

Strengths of this article include:

1. Focus on lighting and circadian patterns as they are associated with equine health and well-being. There is no doubt that this is an important area and in need of more work. The solution that is partially explored in this paper is feasible and makes good sense. There is data here that is one more piece of evidence in the growing literature in this area.

2. The use of well researched and developed LED lighting system is exciting and this manuscript is goo impetus to do more work in this area.

Weaknesses of this article include:

1. The experimental design and potential negative impact of uncontrolled sources of variation make it nearly impossible to come to any clear conclusions.

2. This manuscript is twice as long as it needs to be. Almost every part of the manuscript could be reduced by ½ and in so doing would be come more focused and valuable.

Specific comments:

Title: Why are there 2 titles? Perhaps a requirement, but I prefer the 2nd shorter version.

Page 2, lines 38-40; The final statement here is an over interpretation of the data from a study with many sources of variation. The terms “imply” and “may” indicate that the authors likely feel the same way. Perhaps this study contains evidence that would support more research to better understand these last two points?

Page 3, lines 47-49; There are several places through here where greater specificity would improve the writing and strength of the paper. For example, be clearer about how welfare is compromised or what specifically changed. How is sleeping behavior influenced? Consider improving this specificity throughout.

Page 5, lines 102-114; This long discussion in the introduction is meant to support the value of measuring spontaneous blink rate in horses as an indicator of changes in dopamine transmission. While this connection may exist, it is not established in horses and jumping to using it as a response variable is questionable. I think this could still be included as a response variable, but with much less emphasis on what it my be connected to in regard to dopamine physiology.

Page 5, lines 115-118; Consider listing a clear hypothesis.

Page 6, line 126; As soon as any research is being conducted the procedures should be Category C. If this was categorized as B, that should be reviewed and rectified so as not to cause confusion.

Page 7, line 149; Here and throughout an effort should be made to clearly delineate stalls, from section of the barn from the barn. Is Table 1 a table or a figure? It seems to be more a figure, but it leads to some lack of clarity as to how the sections of the barn were separated. Is the dark line that separates A from B an actual partition?

Page 7, line 159; It is unclear why red light was needed all night in this experiment, particularly as this was not the main focus of the experiment. It would have been better to investigate this in a separate experiment.

Table 5; Is lateral recumbency sleep or awake? Based on later reading, I believe it to be categorized as sleep. For consistency and clarity, that should be listed here as it is for the other variables. In fact it would make sense to define each row with these terms.

Page 15, lines 295-299; The large number of different individuals evaluating video footage is understandable and likely necessary. The use of agreement statistics to evaluate any vaiability or concerns here is good. However, the discussion of the lead author spot checking and a lack of description of how early training and calibration of students was performed leads to some concerns as to the additional noise that may be in this data.

Page 15, line 306; Please clarify the differential choice to use Shapiro Wilkes in one place and D’Agostino & Pearson in another for normality testing.

Page 15, lines 308-310; The description of ANOVA or Friedman’s here is unclear. What data was this conducted upon? What was the reasoning? More details of the models used should be listed. A better description of the model will be important in fully interpreting tables 6 and 7. “repeated measures” does not need to be abbreviated.

Page 16, lines 337-341; This section on ICC and agreement should really be listed in the materials and methods.

Page 17, lines 343-345; The primary sentence her that describes table 6 is unclear. Consider rewording.

Table 6; In this table, the extra time spent out corresponds to extra time spent standing in the stall. These are both in the treatment group and the closest to “significance”. This simply highlights potential source of variation that are outside the primary aims of the experiment.

Table 6; Why is there so much missing data at night? Much more than the day!

Table 7; It is curious that the p-value for the comparison of total recumbency during the day of 458 vs 18 is >0.999, while many other p-values are much lower. Is this correct?

Page 21, line 390; The results here need to be more clearly described. Ultimately, there were day vs night differences. Further is a question of whether there was any interaction between lighting condition and day or night. The way this is written could easily be misinterpreted to indicate that the lighting condition is a “part” of the night versus day effect.

Page 23, line 448; The lack of a treatment effect in this study is not completely surprising, mostly due to the poor control of so many other sources of variation. At the end of the day, this is a poorly setup experiment, so positive or negative conclusions are very difficult to draw.

Page 24, lines 467-479; Far too much discussion of a non-significant difference. Delete all of this.

Page 25, lines 486-488; This is an interesting point and worth attention, particularly to others that may be developing or planning similar work.

Page 25, lines 489-490; While nonsignificant, time outside the stable was on contrast that came closest to significance. Feels a bit as if results are being cherry-picked only if they fit the theories of the authors. In lines 491-492, details are give about activities that were not consistently recorded. This is problematic as it is impossible to really analyze.

Pages 25-26, lines 499-510; Not critical, but again, information that could be helpful to others attempting to complete this kind of work.

Page 26, line 512; “does not negatively influence” is an awkward double negative that may lead to an unclear interpretation. The attention to and reporting of a lack of effect of redlight at night should be removed from the manuscript. It was not a primary objective, and the experimental design does not allow for this analysis appropriately. Really a separate experiment should be conducted. At best, this should be a brief mention.

Page 27, line 530; Rewording required.

Page 27, lines 537-543; This paragraph should be deleted. This analysis goes beyond what the data presented can support.

Pages 27-28, lines 544-563; Delete this paragraph. It is not supported by the results of this study and detracts from a clear and concise manuscript.

Page 28, line 576; “quasi-experimental nature of this study” highlights the major weakness of this study. There is value in what was done here and the data/results should be presented in a much more clear and concise manner for others to use in designing future research in this area.

Page 29, lines 596-597; Due to the fact that there is not a clear definition of what “circadian health” is, id don’t think that the data from this experiment support this statement. The horses in this study were not clearly stressed, but their health and well-being is not 100% clear. Mightn’t they have been healthier if they had more time in paddocks? Just unknown.

Page 29, lines 598-600; None of this is really supported by the findings of this study, particularly any potential benefit of dim red light. More work is needed to support this.

Page 29, lines 602-603; A comparison of wakefulness between the treatment and controlled groups during the day indicates no difference. It is the day vs night difference that is significant. This feels like it is being twisted to support the authors’ hypothesis that the treatment is better.

References; There are too many references. The authors should be more selective. This is not a literature review.

Figure 10; This figure could be improved by bringing the graphs closer together and using a common x-axis.

General comments:

1. There are a number of sources of variation that could have easily been avoided and are disappointing to see. I will record some of those here. The LED lighting system was on a timer, but the control system was turned on and off manually, with greater variation. Some horses were blanketed overnight and some were not and it is unclear who and when and if this had any effect. One horse was fed concentrate meals while the others were not. Another source of variation. The exercise regimes and time out of the stable further contributes to variation. There is lack of clarity as to how this was accounted for in the statistical analysis. There is some talk of the “spillover” of light from the sections or stalls of the barn. The authors seem to indicate that this is not a large concern, and yet their research and much of the other research in this area, that a small exposure to other wavelengths can have a significant effect. All these things combined represent a significant concern in the interpretation of findings from this study.

2. One of the major highlighted findings is the “higher wakefulness during the day in the LED group”. This comparison is somewhat misleading as the difference really seems to lie in the fact that there was lower wakefulness recorded in the LED group at night. This difference does not translate into significant differences in sleep or recumbency. Every effort should be made to more clearly communicate and not potentially mislead here.

3. In the statistical analysis section of the materials and methods, consider not using an entire sentence to describe the software. The analysis performed is most important and the software and version used should simply be listed in parentheses.

4. There are different perspectives on my point here; therefore, a suggestion. Remove the term “significant” except for where it is defined in the statistics section. If anything is deemed “different” or larger or smaller, one would assume that the difference reported was significant, or it was undetectable. In the same vein, either list specific p-values or no p-values, but listing < or >0.05 throughout is not particularly helpful. I would lean towards listing specific p-values.

5. Tables 6 and 7 are foundational to this manuscript, but the differences and reasons to have both tables is unclear. Clearly there were different statistical analyses conducted for each table. Why is table 7 alone not sufficient? It becomes particularly challenging when trying to compare the numbers in the 2 tables, as they don’t really match up, for example total recumbency. The multiple comparisons in table 7 are kind of like an interaction from an ANOVA? I am concerned that these individual comparisons and a lack of a clear correction for these multiple comparisons can lead to an increased risk of Type I errors.

6. There are too many figures illustrating individual horse data from this study. Particularly because most of them are illustrating data without differences. Consider rethinking the most important figures or table that must be presented. I think the number could be reduced by ½ to 2/3rds. There is also not a clear reason to have individual horse data in all of the figures.

7. The general comment #1 above and various points through the paper lead to the conclusion that lack of control is a serious weakness for this manuscript. A potential way forward, to facilitate communication of the data and knowledge captured with this effort, would be to simplify the presentation of data considerably. Focus on table 6 or perhaps table 7 and go no further. Perhaps even a short communication that could be used to spur further research.

6. PLOS authors have the option to publish the peer review history of their article (what does this mean? ). If published, this will include your full peer review and any attached files.

**Do you want your identity to be public for this peer review?** For information about this choice, including consent withdrawal, please see our Privacy Policy .

Reviewer #1: No

Reviewer #2: No

---

## [Author Response · Author response to Decision Letter 1]

16 May 2025

Dear Editor and Reviewers,

We are very grateful for the detailed and thoughtful comments by both reviewers. We have amended our manuscript extensively in light of comments and provide a point-by-point rebuttal (in black font) below. The amendments have significantly improved the clarity, impact and conciseness of the manuscript which we hope you will find is now suitable for publication.

5. Review Comments to the Author

Reviewer #1: Overall comments: An interesting study. I enjoyed reading this manuscript. There are some edits to be made, and those are noted below.

Specific comments:

Line 50-51: please recheck grammar for this sentence, it seems choppy jumping from talking about environmental factors in horses and then light stimuli. Perhaps it needs to be 2 separate sentences?

Thank you. This sentence has been rechecked and improved for clarity.

Line 57-60: need an additional comma after hypothalamus

This edit has been made.

Line 76-77: tie back in the connect to melatonin in the final sentence of this paragraph.

Thank you. We have tied this point back in as follows: “Therefore, consistent sleep-wake cycles and healthy functioning of the circadian system requires the absence of light at night to facilitate the normal rise in melatonin production [30]”.

Line 78-81: what about melatonin? The light type that stimulates the circadian clock is discussed, but the way the sentence structure is set up, it feels like it is currently lacking a mention of the optimal stimulation for melatonin. OR – rephrase the sentence. Perhaps the part after the semicolon should just be its own sentence.

Thank you. We have made the section after the semicolon its own sentence, improving the clarity. Melatonin regulation by light is then addressed further on in the paragraph.

Line 97-101: is there research in any other mammalian species on red light and its impact on melatonin and the circadian rhythm?

There is in rodents, white light intensities as low as 5 lux have been shown to disrupt the sleep-wake cycle (Stenvers et al., 2016) whereas <10 lux of red light was shown to facilitate normal rest-activity behaviour and metabolic function (Zhang et al., 2017; Opperhuizen et al., 2017). We have now included these additional information and citations here.

Line 131-133: please provide number of mares and geldings, as well as breed.

The numbers of geldings and mares have been added. Specific breed details are not available other than to say they were mixed riding horse/cob breed types.

Line 142-143: what type of forage? Be more specific

The forage type has now been specified as hay.

Line 135: do you mean the stalls were partitioned by wooden panels? Not the stables?

Stalls (US) are termed ‘stables’ in the UK. In this line we state the size of individual stables within the larger barn that they are housed within.

Line 152: Table 1 – just to clarify – the treatments were two separate barns? Or the same barn? It is unclear from this picture. If it is two separate barns, I would label the table to clarify.

The treatments were in the same barn with both lighting treatments installed in each individual stable. This information is detailed beneath the table. This arrangement avoided any requirement to move horses to a different stable for the second lighting treatment, avoiding unnecessary disruption and the need for additional acclimatisation time. The caption has been amended as follows to improve clarity: “Schematic of the experimental barn layout and horse assignment to individual stables and groups (A and B) for cross-over design.”

Line 187 and line 188: barn and stable are used here interchangeably. Please pick one term to use and correct throughout paper.

Thank you. We have corrected this. ‘Stable’ is used for the individual horse enclosure and ‘barn’ used to describe the building the stables are housed within.

Line 194-195: So there was no washout period in between the two trials? Is there literature to show this has no effect on variables being measured?

There was no requirement for a wash-out period between the two lighting treatments as the control lighting represented the normal lighting regime that the horses were accustomed to. A 4-week experimental period was chosen as it can take two weeks for circadian rhythms to fully adapt to a new photoperiod in horses (Murphy et al., 2007). However, even though there was little change in photoperiod between treatments for this study, all that changed was the intensity and spectral quality of the light, this buffer was allowed for and extended to ensure any and all behavioural parameters had fully adapted to the new lighting. The variables were then measured during week 4 of each treatment.

Line 200: I don’t think table 3 is really necessary. The protocol is explained sufficiently in the prior paragraph.

Thank you. We agree and have removed Table 3.

Line 347: consider moving the text in row 3 ( duration of behaviors in sec) to the bottom of the table – it’s a little jarring where it is now. And why for the results in this table are then H shown in all the rows, but not sec? I would either include both or neither, since you explain the units in row 3.

Respectfully, we feel it is best to define the units above the columns. However, we have shortened the description in row 3 so it looks less busy. We have also now removed the reference to hours (h) in all the rows as you correctly point out that these have previously been defined in row 3.

Line 438: was there any effect of age on the horses on behavior and arousal?

This effect was not examined as all horses bar one (aged 17) were aged from 7 to 15 years, classed as mature horses, not young (<7) or aged (>15). Plus, the groups were blocked for age to avoid any age-related confounding factors.

Line 489-498: Frankly, I do not this paragraph is necessary. There was no significant effect found by the researchers on time spent outside, and this did not seem to be a primary point of interest for the study. Especially considering the horses did different things when outside. While I agree looking at exercise and sleep patterns would be interesting, given the length of this paper, and sheer speculative manner of this particular paragraph, I would cut it.

Thank you. We agree and have now removed this entire paragraph from the manuscript.

Figure 2: are there better-quality images for this figure? The ones provided are a bit blurry.

The Figures uploaded on submission were high quality and did not appear blurry, but it is quite likely that they may only appear blurry on the generated pdf for review. We have checked that all Figures again again meet with journal specifications.

Figure 3: were any pre-trials done to see if the location of the go-pro impacted horses lying down to sleep? It seems based on this picture, if the go-pro happened to be on the side the horse preferred to be on when lying down, that this could skew results.

The horses were fitted with the go-pro attachment to record SBR for a 30-min duration only, during which time they were continuously observed while standing and did not lie down.

Figure 6-10: again, the resolution of these images seems a little blurry. Would recheck to make sure the best quality is uploaded for the manuscript.

Thank you. Each of these will be rechecked but we believe it is a result of the PDF generation

Reviewer #2: Review of “Influence of lighting on sleep behaviour, circadian rhythm and spontaneous blink rate in stabled riding school horses (Equus caballus)”.

Overview: This manuscript describes a study in which horses were exposed to 2 lighting sources and a host of behavioral and physiologic response variables were evaluated. The results indicate that in general there were not detectable differences between the treatment and control groups. There were clear differences in behavior between day and night and there was good evidence for circadian rhythms. This manuscript could be improved by focusing on the clear and obvious findings. It is likely that a manuscript that was 1/3 to ½ the length would be much more clear and concise and be of value to those that are conducting research in this area.

Strengths of this article include:

1. Focus on lighting and circadian patterns as they are associated with equine health and well-being. There is no doubt that this is an important area and in need of more work. The solution that is partially explored in this paper is feasible and makes good sense. There is data here that is one more piece of evidence in the growing literature in this area.

2. The use of well researched and developed LED lighting system is exciting and this manuscript is goo impetus to do more work in this area.

We thank the reviewer for highlighting some strengths of the manuscript.

Weaknesses of this article include:

1. The experimental design and potential negative impact of uncontrolled sources of variation make it nearly impossible to come to any clear conclusions.

Respectfully, the crossover experimental design employed in this experiment was specifically chosen due to its ability to significantly reduce sources of variation, the primary one being individual variation. This type of design eliminates or reduces the impact of individual variability, which can be a significant source of noise in between-subjects designs (like parallel group trials). As each horse acted as its own control and received both treatments, the power of the study increased significantly and equated to a similar study of > 20 horses without a crossover (parallel group trial). The design also accounted for a potential effect of time which creates additional potential for variation in a longitudinal study. Furthermore, this choice of design allowed for a reduced sample size, while increasing the precision of the estimated treatment effect. Reducing sample size in this manner is an important component of ensuring adherence to the three R’s, namely Replacement, Reduction, and Refinement, the principles that guide humane animal research. We will hope to further convince the reviewer of how sources of variation were controlled for in responding to subsequent comments.

2. This manuscript is twice as long as it needs to be. Almost every part of the manuscript could be reduced by ½ and in so doing would be come more focused and valuable.

We appreciate that the manuscript is long and will endeavour to reduce its length substantially in response to both reviewer comments.

Specific comments:

Title: Why are there 2 titles? Perhaps a requirement, but I prefer the 2nd shorter version.

This is a requirement of the journal.

Page 2, lines 38-40; The final statement here is an over interpretation of the data from a study with many sources of variation. The terms “imply” and “may” indicate that the authors likely feel the same way. Perhaps this study contains evidence that would support more research to better understand these last two points?

Respectfully, we do strongly feel that that our findings imply the safe use of dim red light at night over stabled horses for maintenance of normal sleep behaviour and we hope to convince the reviewer further of this within our rebuttal. However, we appreciate the direction on wording in relation to the finding related to wakefulness and have amended the final part of the abstract to read “and provides evidence supporting further research to better understand the role of blue-enriched LED light at promoting increased wakefulness during daytime in stabled horses.”

Page 3, lines 47-49; There are several places through here where greater specificity would improve the writing and strength of the paper. For example, be clearer about how welfare is compromised or what specifically changed. How is sleeping behavior influenced? Consider improving this specificity throughout.

Thank you. The specificity in these lines has now been improved with specific examples from the cited literature describing how welfare is compromised and how sleep behaviour is influenced.

Page 5, lines 102-114; This long discussion in the introduction is meant to support the value of measuring spontaneous blink rate in horses as an indicator of changes in dopamine transmission. While this connection may exist, it is not established in horses and jumping to using it as a response variable is questionable. I think this could still be included as a response variable, but with much less emphasis on what it my be connected to in regard to dopamine physiology.

Using dopamine agonist administration protocols, the link between elevated dopamine and blink rate has been tested in mammalian species with relatively primitive brains (rodents) (Kaminer et al., 2011) and more developed equivalents (non-human primates) (Kleven and Koek, 1996). The horse, falling between these two examples, is highly unlikely to exhibit differential characteristics in respect of dopamine and blink rate. However, we accept that to date, this has not been empirically tested. Therefore, we have shifted the emphasis away from dopamine transmission, over to what blink rate is telling us about the animal’s response to stress. In this regard, Mott and McBride (2020) recorded a significant positive correlation between salivary cortisol and blink rate in the horse (r = 0.56, p < 0.001), suggesting that blink rate can indeed be used as an indication of stress perception in this species. The introduction has been amended to reflect this change.

Page 5, lines 115-118; Consider listing a clear hypothesis.

Thank you. We have now added a clear hypothesis statement.

Page 6, line 126; As soon as any research is being conducted the procedures should be Category C. If this was categorized as B, that should be reviewed and rectified so as not to cause confusion.

We gained funding from the Morris Animal Foundation who, as part of the funding bid, ask extensively about the ethics and welfare of the study and for categorisation against USDA pain and distress categories. This information is scrutinised by their Animal Welfare Advisory Board and no concerns were raised about the use of the Category B description during this process. We refer the reviewer to the categorization resource https://www.morrisanimalfoundation.org/sites/default/files/files/2018-12/USDA-Pain-and-Distress-Categories.pdf . We referred to the procedures within our project as being within routine husbandry procedures including physical restraint and where we did not manipulate the environment in order to observe animal behaviour.

Page 7, line 149; Here and throughout an effort should be made to clearly delineate stalls, from section of the barn from the barn. Is Table 1 a table or a figure? It seems to be more a figure, but it leads to some lack of clarity as to how the sections of the barn were separated. Is the dark line that separates A from B an actual partition?

A ‘stall’ is referred to as a ‘stable’ in the UK. The individual stables were housed within an American barn style building. Edits have been made throughout to make this more clear in response to both reviewers. The reviewer is correct that Table 1 is a schematic and is more suitable to be labelled and included as a Figure, which has now been done. The dark line across the aisleway has been removed as there was no additional partition here. Wooden panel walls that did not extend completely to the ceiling divided the stables. This is detailed further down in the Methods.

Page 7, line 159; It is unclear why red light was needed all night in this experiment, particularly as this was not the main focus of the experiment. It would have been better to investigate this in a separate experiment.

We are grateful that the reviewer has highlighted this lack of clarity. The aim of this experiment was to compare a commercially available lighting system (a system the reviewer commended earlier as a “well researched and developed LED lighting system”) designed to optimise circadian rhythms in the horse

---

## [Editor Report · Decision Letter 1]

Influence of lighting on sleep behaviour, circadian rhythm and spontaneous blink rate in stabled riding school horses (Equus caballus)

PONE-D-24-58876R1

Dear Dr. Murphy,

We’re pleased to inform you that your manuscript has been judged scientifically suitable for publication and will be formally accepted for publication once it meets all outstanding technical requirements.

Kind regards,

Paul A. Bartell

Academic Editor

PLOS ONE
---

## [Editor Report · Acceptance letter]

PONE-D-24-58876R1

PLOS ONE

Dear Dr. Murphy,

I'm pleased to inform you that your manuscript has been deemed suitable for publication in PLOS ONE. Congratulations! Your manuscript is now being handed over to our production team.

Kind regards,

on behalf of

Dr. Paul A. Bartell

Academic Editor

PLOS ONE